# Independent genomic polymorphisms in the PknH serine threonine kinase locus during evolution of the *Mycobacterium tuberculosis* Complex affect virulence and host preference

Elena Mata[1,2], Damien Farrell[3], Ruoyao Ma[3], Santiago Uranga[1,2], Ana Belen Gomez[1,2], Marta Monzon[4], Juan Badiola[4], Alberto Anel[5], Jesús Gonzalo-Asensio[1,2,6], Carlos Martin[1,2,7], Stephen V. Gordon[3‡], Nacho Aguilo[1,2‡]*

1 Grupo de Genética de Micobacterias, Universidad de Zaragoza/ISS Aragon, Zaragoza, Spain, 2 CIBER Enfermedades Respiratorias, Instituto de Salud Carlos III, Madrid, Spain, 3 School of Veterinary Medicine, Veterinary Science Centre, University College Dublin, Dublin, Ireland, 4 Research Centre for Encephalopathies and Transmissible Emerging Diseases, Universidad de Zaragoza, Zaragoza, Spain, 5 Grupo Apoptosis, Inmunidad y Cáncer, IIS Aragón. Dpto. Bioquímica y Biología Molecular y Celular, Fac. Ciencias, Universidad de Zaragoza, Zaragoza, Spain, 6 Instituto de Biocomputación y Física de Sistemas Complejos (BIFI), Zaragoza, Spain, 7 Servicio de Microbiología, Hospital Universitario Miguel Servet, ISS Aragon, Spain

‡ These authors are joint senior authors on this work.
* naguilo@unizar.es

## Abstract

Species belonging to the *Mycobacterium tuberculosis* Complex (MTBC) show more than 99% genetic identity but exhibit distinct host preference and virulence. The molecular genetic changes that underly host specificity and infection phenotype within MTBC members have not been fully elucidated. Here, we analysed RD900 genomic region across MTBC members using whole genome sequences from 60 different MTBC strains so as to determine its role in the context of MTBC evolutionary history. The RD900 region comprises two homologous genes, *pknH1* and *pknH2*, encoding a serine/threonine protein kinase PknH flanking the *tbd2* gene. Our analysis revealed that RD900 has been independently lost in different MTBC lineages and different strains, resulting in the generation of a single *pknH* gene. Importantly, all the analysed *M. bovis* and *M. caprae* strains carry a conserved deletion within a proline rich-region of *pknH*, independent of the presence or absence of RD900. We hypothesized that deletion of *pknH* proline rich-region in *M. bovis* may affect PknH function, having a potential role in its virulence and evolutionary adaptation. To explore this hypothesis, we constructed two *M. bovis* 'knock-in' strains containing the *M. tuberculosis pknH* gene. Evaluation of their virulence phenotype in mice revealed a reduced virulence of both *M. bovis* knock-in strains compared to the wild type, suggesting that PknH plays an important role in the differential virulence phenotype of *M. bovis* vs *M. tuberculosis*.

**Data Availability Statement:** The raw sequencing data are available in the Sequence Read Archive (SRA) under the bioproject ID PRJNA645286.

**Funding:** CM and NA received a grant from Spanish Ministry of "Ciencia, Innovación y Universidades" (www.ciencia.gob.es/portal/site/MICINN/) [grant number RTI2018-097625-B-I00]. SVG acknowledges funding from Science Foundation Ireland (www.sfi.ie/) (grant number 15/IA/3154). RM was a recipient of a China Scholarship Council studentship (www.chinesescholarshipcouncil.com/). The funders had no role in study design, data collection and analysis, decision to publish, or preparation of the manuscript.

**Competing interests:** The authors have declared that no competing interests exist.

## Author summary

Tuberculosis is caused in humans and animals by organisms from the *Mycobacterium tuberculosis* Complex (MTBC), that share more than 99% genetic identity but exhibit distinct host preference and virulence. While *Mycobacterium tuberculosis* is the main causative agent of human TB, *Mycobacterium bovis* is responsible for bovine TB disease, that exacts a tremendous economic burden worldwide, as well as being a zoonotic threat. Unlike the human restriction of *M. tuberculosis*, *M. bovis* has a broader host range and it has been found to be more virulent than *M. tuberculosis* in different animal models. However, the molecular basis for host preference and virulence divergence between *M. tuberculosis* and *M. bovis* is not fully elucidated. Here we study the genetic variations of the genomic region RD900 in the context of MTBC phylogeny. RD900 contains two genes encoding orthologues of the serine/threonine kinase PknH, which is linked to the regulation of several bacterial processes including virulence. We found that *M. bovis* *pknH* genes show a conserved deletion that is not present in *M. tuberculosis* strains, and we evaluated the potential impact of these variations in the regulation of *M. bovis* vs *M. tuberculosis* virulence through the construction and *in vivo* characterization of *M. bovis* *pknH* mutant strains.

## Introduction

The *Mycobacterium tuberculosis* Complex (MTBC) is composed of several highly genetically related mycobacterial species (>99% nucleotide sequence identity) that infect and cause tuberculosis (TB) in a range of mammalian hosts. MTBC species can be broadly classified into human-adapted mycobacteria, comprising the obligate human pathogens *Mycobacterium tuberculosis* sensu stricto and *Mycobacterium africanum*, and animal-adapted mycobacteria, referring to those that can propagate and transmit in a range of wild and domesticated animal hosts. Despite their high degree of genomic identity, MTBC members exhibit important differences in relation to host range, transmissibility, pathogenesis and virulence [1, 2].

*Mycobacterium bovis* is the causative agent of bovine TB, which is responsible for high economic losses in livestock productivity and disease control costs and remains a major problem in many developed livestock-producing countries as well as in developing countries [3]. Unlike the human restriction of *M. tuberculosis*, *M. bovis* has a broader host range. Although it mainly sustains infection in cattle, *M. bovis* infection affects many other mammalian species and can be found in a range of maintenance hosts such as deer, possums and badgers that act as wildlife reservoirs. The route of transmission from these reservoir species to cattle and vice versa is not fully elucidated, but it is proposed that close contact, aerosol and faecal-oral routes play a role [3, 4]. Furthermore, *M. bovis* poses a risk as a zoonosis for humans, mainly through the consumption of raw milk or close contact with infected cattle. However, transmission of *M. bovis* among immunocompetent human hosts is uncommon, representing a dead end or spillover host [5]. Conversely, *M. tuberculosis* has been independently shown to be attenuated in cattle [6, 7], while *M. bovis* strains show greater virulence than *M. tuberculosis* in animal models such as mice or rabbits [8–10].

*M. tuberculosis* has evolved an extensive signalling network that includes, among others, 11 serine threonine protein kinases (STPKs) involved in the regulation of many aspects of bacterial physiology. STPKs have been shown to play a crucial role in the growth and survival of *M. tuberculosis* during infection, controlling a variety of cellular processes related to cell division, cell wall biosynthesis, cellular metabolism, transcriptional regulation and virulence [11]. PknH is a transmembrane STPK that presents a typical conserved N-terminal kinase domain

followed by a proline-rich region, a predicted transmembrane single helix domain and an extracellular C-terminal sensor domain. It has been shown that PknH phosphorylates itself and other substrates, and it plays a role in cell wall biosynthesis, response to nitric oxide stress and the dormancy response [12–15]. PknH has also been implicated in virulence, as an *M. tuberculosis* H37Rv *pknH* knock-out mutant was shown to be hypervirulent in a mouse model [16].

The organization of the genome region that contains the *pknH* gene exhibits substantial variation across MTBC strains. The *M. tuberculosis* H37Rv genome contains a single copy of the *pknH* gene which is located downstream of *embR*, one of the substrates phosphorylated by PknH [12]. Genome sequencing of the *M. africanum* GM041182 strain revealed a different genetic organization to that observed in *M. tuberculosis* H37Rv, showing two different *pknH* genes (*pknH1* and *pknH2*) that flank a potential ABC transporter named *tbd2*. This observation led to the description of a new Region of Difference, named RD900, that was first described as a lineage specific locus. The RD900 locus is 3141 bp long and contains a single complete gene (MAF12860, *tbd2*) and the 3' end of another (MAF12870, *pknH2*) [17]. This region was initially thought to be deleted in *M. bovis* and *M. tuberculosis* "modern" lineages. However, although RD900 locus was not present in the original *M. bovis* AF2122/97 genome annotation, resequencing of this genome revealed that the RD900 locus is actually present in this *M. bovis* strain [18].

Both *pknH1* and *pknH2* genes present in *M. africanum* GM041182 and *M. bovis* AF2122/97 exhibit a high degree of identity between their kinase domains, but they have different sensor domains; this may affect sensing of environment signals by PknH1 and PknH2, and consequently may result in differential cellular responses. The proline-rich region of *M. africanum* *pknH2* has a deletion of about 100 bp compared to *pknH1* and to the *M. tuberculosis* H37Rv *pknH* orthologue. The same proline-rich region is deleted in *M. bovis* AF2122/97 *pknH2*, but is also deleted in *pknH1* [17, 18].

Here, we analyzed the RD900 genomic region across MTBC members to determine its plasticity across MTBC evolutionary history, using whole genome sequencing data from 60 representative MTBC strains. In addition, we performed a comparative study between *M. tuberculosis*, *M. bovis* and *M. africanum* strains regarding their virulence and dissemination capacity in mice. We also hypothesized that the PknH proline rich-region deletion found in *M. bovis* could have an impact on the enhanced virulence associated with these strains. To elucidate this hypothesis, we constructed two *M. bovis* strains in which we introduced the *M. tuberculosis* *pknH1* gene, and determined their virulence phenotype in mice. We finally used transcriptomics to assess the impact of introduction of the *M. tuberculosis* *pknH1* gene on global gene expression in the recombinant *M. bovis*.

## Results

### Independent RD900 deletion events across MTBC and a conserved deletion in *M. bovis* and *M. caprae pknH* genes

In order to delineate variation in the RD900 locus and *pknH* genes across the different MTBC lineages, and the potential evolutionary impact of this variation, we performed an analysis of this genomic region using WGS data from 60 isolates representative of the different MTBC lineages (S1 Table). The original sequence reads of the different MTBC strains were downloaded in fastq format from GenBank and aligned to the reference sequence of *M. africanum* GM041182 RD900 locus, including the *tbd2* gene, both *pknH* genes (*pknH1* and *pknH2*) and the two flanking genes *embR* and *MAF_RS06695*. The resulting alignments were first analysed

for the presence or absence of RD900, and for the presence or absence of the deletion within the proline-rich region in the *pknH* genes.

The RD900 region was found to be present in all analysed genomes of *M. canettii*, a 'smooth' colony species thought to be similar to the progenitor of the MTBC. The *M. canettii* genome shows the same genomic organization as *M. africanum*: the *tbd2* gene flanked by two *pknH* genes and a deletion within the proline-rich region of *pknH2* but not *pknH1* (Fig 1A). The same locus organization was present in the analysed *M. tuberculosis* Lineage 7 isolates. However, the presence of RD900 was found to be variable among *M. tuberculosis* Lineage 1 strains. Some of the analysed isolates contained an intact RD900 region whereas others had a single complete *pknH* gene containing an intact proline region sequence. As previously described, RD900 was found to be deleted in all the analysed isolates from *M. tuberculosis* "modern" lineages (Lineages 2, 3 and 4) which are distributed worldwide. In contrast, RD900 was found to be present in all the analysed *M. africanum* Lineage 5 and Lineage 6 isolates, although variability was observed within the proline-rich region sequence of *pknH1* which was intact in some isolates and deleted in others.

In the animal adapted species, the RD900 region was found to be present in the analysed isolates from *M. mungi*, *M. microti* and *M. pinnipedii*, with the sequence encoding the proline-rich region deleted in *pknH2* but not *pknH1*. In contrast, RD900 was found to be deleted in *M. orygis*, which presented a single *pknH* gene containing the complete proline-rich encoding region. Finally, the presence of RD900 in *M. caprae*, *M. bovis* and BCG strains was found to be variable among the different isolates analysed. In all of these latter strains, both with or without the RD900 region, we found that the different *pknH* gene copies had the proline-rich region deleted. The only exception came from one *M. bovis* strain (*M. bovis* B2 7505) in which RD900 was deleted and the resulting single copy of *pknH* had a complete proline rich region. This strain was isolated from a human patient in Uganda [19] and was subsequently described to show *M. tuberculosis* RD patterns [20]; hence, it appears actually to be an *M. tuberculosis* strain.

In order to better understand the evolutionary process that this particular region underwent across the MTBC members, we analysed the presence of point mutations (synonymous, missense and frameshift mutations) in the RD900 locus across the analysed strains through a variant and consequence calling analysis. The results for the missense and frameshift mutations are represented in a clustered heat map showing the presence/absence data for each mutation (Fig 1B). Mutations present in only a single isolate from the same lineage were excluded. In the clustered heatmap it can be observed that some isolates of the same lineage are clustered together whereas other isolates from the same lineage are not clustered, reflecting the inter-strain variability of this region across the different MTBC lineages. The *M. bovis* and *M. caprae* strains with an intact RD900 region were more closely related than those strains in which the RD900 region was deleted, reflecting their different evolutionary pathways.

Finally, in order to confirm the results obtained from the genomic analysis of *pknH* genes, and focusing on the differences between *M. tuberculosis* and *M. bovis* strains, we designed specific primers to verify the presence or absence of the proline-rich region in *M. bovis* *pknH* genes compared to *M. tuberculosis*. To confirm the presence of this deletion, a PCR was performed using a primer pair flanking the region of the deletion. This PCR was performed using DNA extracts from *M. tuberculosis* H37Rv and different *M. bovis* strains: the reference strains AF2122/97 and AN5; the multidrug resistant strain *M. bovis* B, responsible for large TB outbreaks in Spain [21]; and a BCG strain. An amplification product of lower molecular weight was obtained from the DNA extracts of all the tested *M. bovis* strains compared to *M. tuberculosis* H37Rv, reflecting the deletion of the proline-rich region in *M. bovis* strains (Fig 1C).

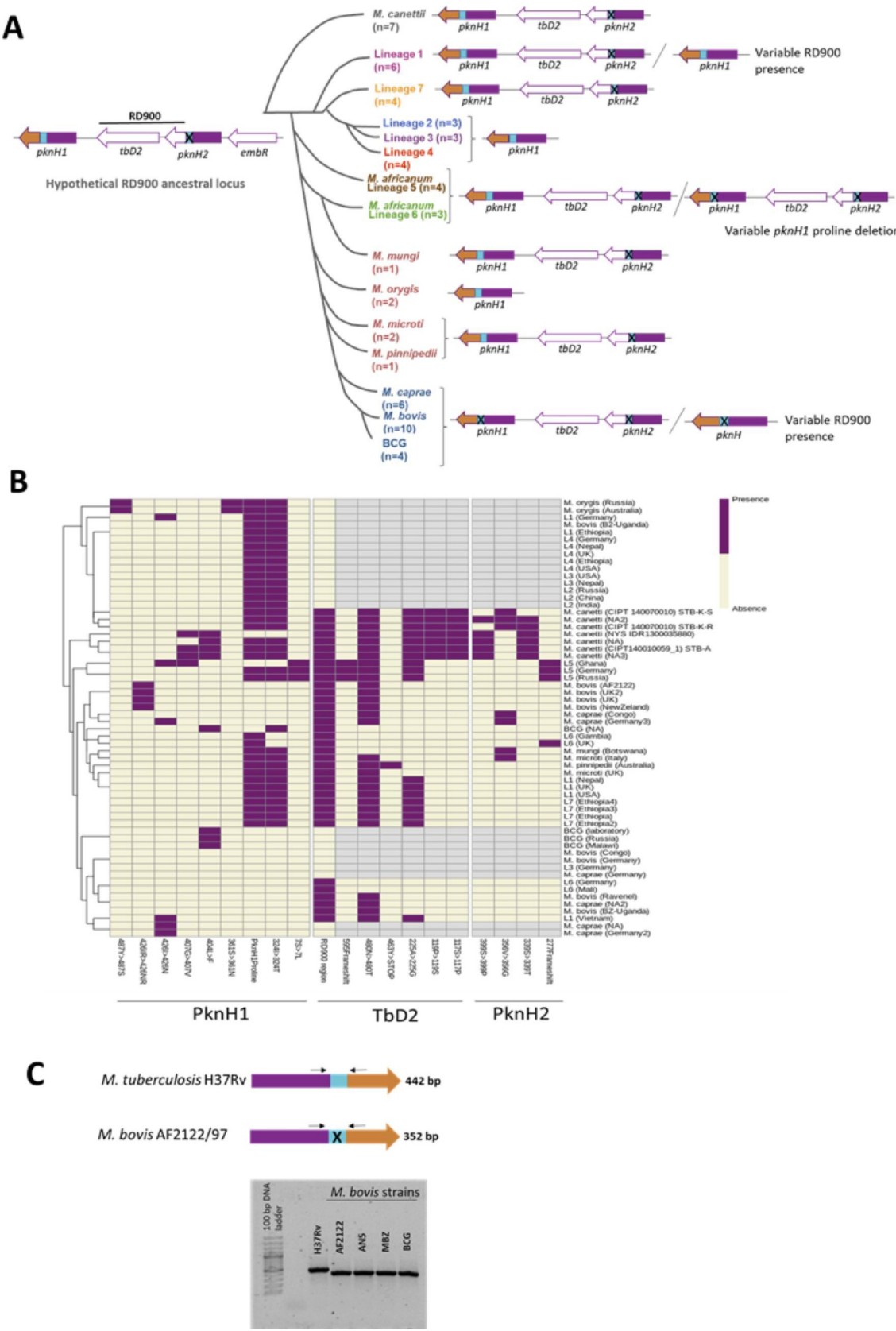

**Fig 1. RD900 analysis across MTBC members.** (A) Schematic representation of RD900 locus organization in MTBC. *pknH* genes are represented with the kinase domain in purple, the proline rich region in blue and the different sensor domain in orange for *pknH1* and white for *pknH2*. The presence of the deletion in the proline rich region is represented with an "X". The number of analysed isolates (n) for each MTBC specie is indicated. (B) Presence/absence of missense and frameshift mutations in RD900 locus across MTBC. Clustered heat map constructed from the presence/absence data of missense and frameshift mutations in RD900 locus from the MTBC analysed isolates. The presence or absence of RD900 and the deletion within the proline-rich region in PknH1 is also indicated. (C) Verification of the presence of proline-rich region deletion in *M. bovis* AF2122/97, AN5, MBZ and BCG and *M. tuberculosis* H37Rv *pknH* genes by PCR using a primer pair flanking the deleted region.

## *M. bovis* disseminates more efficiently than *M. tuberculosis* and *M. africanum* in a mouse model

We next evaluated *in vivo* infection and dissemination differences between MTBC members by performing a comparative virulence experiment in mice using *M. tuberculosis*, *M. bovis* and *M. africanum* strains. C57BL/6 mice were infected by the intranasal route with a low dose challenge ($\sim$ 200 CFUs). We used GFP-expressing strains so that we could assess cell-to-cell spread by flow cytometry. The mycobacterial strains used were the *M. tuberculosis* reference laboratory strain H37Rv, the *M. tuberculosis* clinical isolate Mt103 [22], the *M. bovis* reference strain AF2122/97 and the *M. africanum* clinical isolate HCU2828 (Fig 2A).

Our results showed a higher bacterial load in lungs from mice infected with *M. bovis* compared to those infected with *M. tuberculosis* or *M. africanum* strains. This was observed both at one and four months post-challenge (Fig 2B). Accordingly, lung histopathology analysis revealed more extensive inflammatory foci with inflammatory infiltrate in lung parenchyma along with tissue degeneration in the *M. bovis*-infected group compared to other groups (Fig 2C). We also assessed bacterial dissemination into spleen, liver and kidneys. We found a significantly higher bacillary load in kidneys at both time points in *M. bovis* compared to *M. tuberculosis* and *M. africanum* infected mice. In the case of the liver, significantly higher *M. bovis* dissemination was evident at 4 months post-infection. In general, these results suggested a greater ability of *M. bovis* to disseminate and persist in these non-lymphoid organs. No significant differences were observed in CFU counts from spleen, although bacillary loads were slightly higher in *M. bovis* infected mice (Fig 2D).

We next studied the ability of the different MTBC species to disseminate cell-to-cell within the lung. We analysed by flow cytometry the percentages of GFP-positive cells in cellular suspensions from lungs. At one month post-infection, the proportion of GFP+ cells was found to be higher in the *M. bovis* group, which reflected a greater ability of *M. bovis* to establish at the primary site of infection than *M. tuberculosis* or *M. africanum* strains (Fig 2E and 2F). Total and GFP+ infected lung macrophage and neutrophil populations were analysed using specific surface markers for each population at one month post-infection. In accordance with the enhanced virulence phenotype of the *M. bovis* strain, we found a strongly increased neutrophil influx in this group (Fig 2G and 2H). Interestingly, different studies have shown an association, both in humans and animal models, between an increase of neutrophil infiltration in the lungs and a higher degree of TB disease [23–26].

## Introduction of the *M. tuberculosis pknH* allele into *M. bovis* results in lower virulence in mice

Our results indicated that all the MTBC species, except *M. bovis* and *M. caprae*, contained a *pknH* gene that included the proline-rich region. Thus, considering our findings showing the higher virulence of *M. bovis*, together with a previous study that described a more virulent phenotype of a *pknH* knock-out *M. tuberculosis* H37Rv strain [16], we hypothesized that the

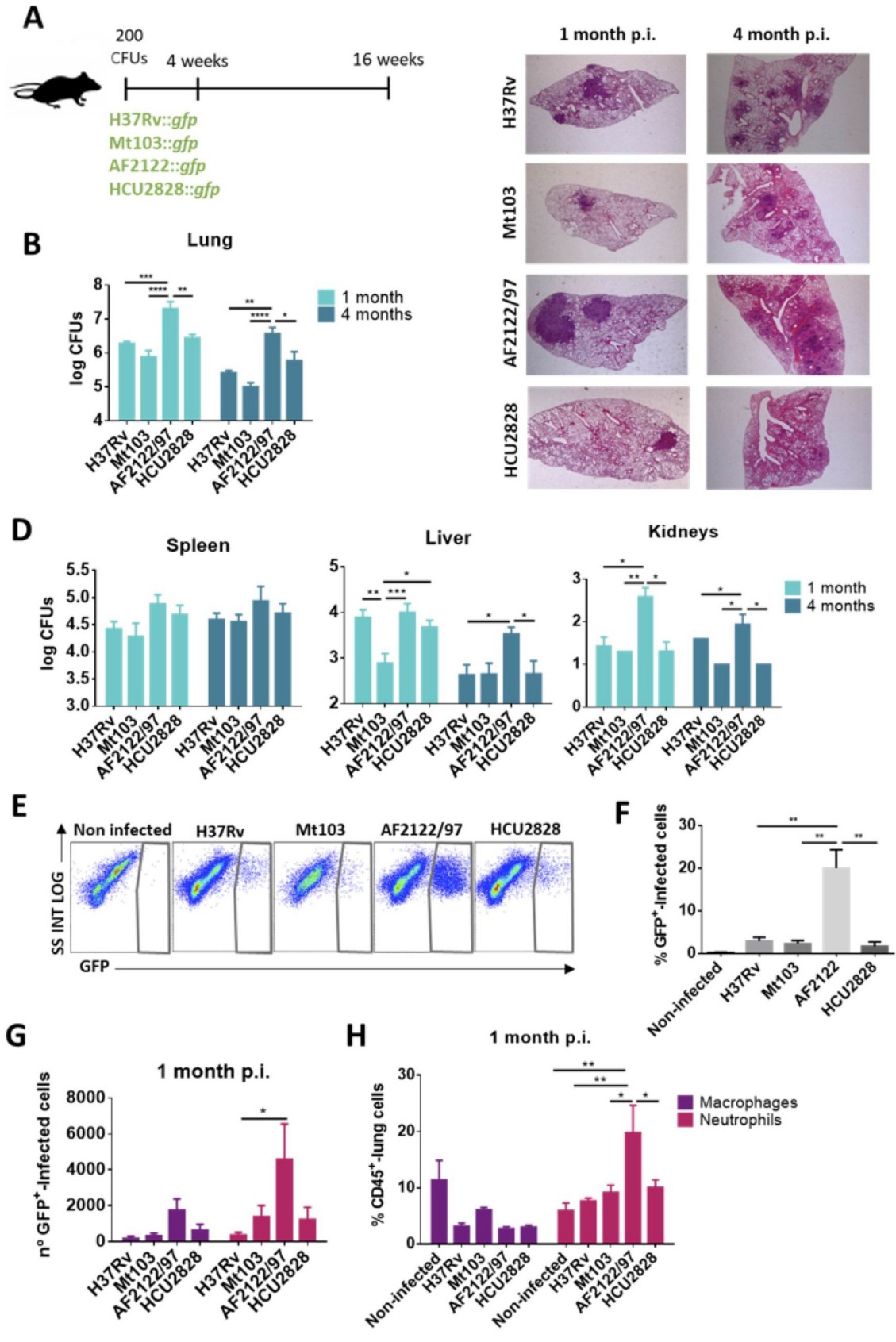

**Fig 2. *M. bovis* exhibits greater virulence than *M. tuberculosis* and *M. africanum* in the mouse model.** (A) Schematic representation of the infection model in C57BL/6 mice. (B) Bacterial burden in lungs at one at four months post-infection. (C) Representative images (1.5x) of heamtoxilin-eosin staining of lungs from mice infected with the different pathogens at one and four months post-infection. (D) Bacterial burden in spleen, liver and kidneys at one and four months post-infection. (E) Representative dot-plots of GFP+-infected lung cells from lung cell suspensions of *M. tuberculosis*, *M. bovis* and *M. africanum* infected mice at one month post-infection. (F) Percentages of GFP+-infected lung cells from lung cell suspensions of *M. tuberculosis*, *M. bovis* and *M. africanum* infected mice at one month post-infection. (G) Cell number of GFP+-infected lung macrophages and neutrophils from lung cell suspensions of *M. tuberculosis*, *M. bovis* and *M. africanum* infected mice at one month post-infection. (H) Percentages of total CD45+ macrophages and neutrophils from lung cell suspensions of *M. tuberculosis*, *M. bovis* and *M. africanum* infected mice at one month post-infection. Data are represented as mean ± SEM from two independent experiments (n = 6 mice/group). Statistical analysis was performed by two-way ANOVA (B,D), one-way ANOVA (F) and multiple unpaired T-test (G,H,I). $^{*}p < 0.05$; $^{**}p < 0.01$; $^{***}p < 0.001$; $^{****}p < 0.0001$.

deletion within the proline-rich region of *M. bovis pknH* genes may lead to an inactive or differently regulated protein that in turn affects virulence.

To address this hypothesis, we constructed two *M. bovis* strains in which the *pknH* gene from *M. tuberculosis* H37Rv was introduced into the chromosome using the integrative plasmid pMV361. The complemented *M. bovis* strains were the reference strain AF2122/91 and the AN5 strain. Both of them are RD900+ *M. bovis* strains, and therefore carry two homologous *pknH* copies, deleted for the proline-rich region (Fig 1C). To achieve comparable levels of *pknH* expression to those observed in H37Rv, the *pknH* gene was cloned under the control of its own promoter.

The introduction of the pMV361 plasmid carrying the *M. tuberculosis* H37Rv *pknH* gene ($pknH^{TB}$) into the *M. bovis* strains was verified by colony PCR (Fig 3A). Using a primer pair flaking the proline-rich region deletion, the recombinant colonies showed two amplification products, one corresponding to the introduced complete *pknH* from *M. tuberculosis* H37Rv (442 bp) and the other one corresponding to the *M. bovis pknH* genes (352 bp). Once it was confirmed that the $pknH^{TB}$ gene was correctly integrated into the knock-in *M. bovis* strains (AF2122::$pknH^{TB}$ and AN5::$pknH^{TB}$), $pknH^{TB}$ expression was evaluated and showed comparable levels in both *M. bovis* knock-in strains and *M. tuberculosis* H37Rv (S1 Fig).

Then, C57BL/6 mice were i.n. infected with a low dose (∼200 CFUs) of the *M. bovis* AF2122/97 and AN5 WT strains and their corresponding $pknH^{TB}$ knock-in strains. Lung histopathology and bacterial burden determination in lungs and spleen were performed after 4 weeks of infection (Fig 3B). A significant reduction in bacterial load in lungs and spleen was observed for both AF2122::$pknH^{TB}$ and AN5::$pknH^{TB}$ strains compared to their wild type *M. bovis* parental strains (Fig 3C). In accordance with expectations, histopathology revealed lower inflammatory damage and tissue degeneration in lungs from the $pknH^{TB}$ knock-in strains (Fig 3D).

To further characterize the phenotype of *M. bovis::$pknH^{TB}$ in vivo*, we constructed a GFP-expressing version of the strain, and we thereafter infected mice with these strains. We first confirmed that lung bacterial burden profile was similar to that observed with non-fluorescent strains at one month post challenge, to check that introduction of the new plasmid did not alter strain virulence (Fig 3F). Analyses of lung cellular suspensions revealed a significantly higher percentage of GFP+ infected lung cells in *M. bovis* AF2122 infected mice than in *M. bovis* AF2122::$pknH^{TB}$ and *M. tuberculosis* H37Rv infected mice (Fig 3G). The analysis of lung infected populations also showed a higher percentage of infected neutrophils in mice infected with the AF2122 wild-type strain (Fig 3H). In this experiment we evaluated antigen-specific IFNγ responses following *ex vivo* stimulation of lung cells with a preparation of tuberculosis antigens (Purified Protein Derivative, PPD). We found a stronger response in cellular suspensions from *M. bovis* AF2122-infected mice compared to *M. bovis pknH^{TB}* knock-in and *M. tuberculosis* groups (Fig 3I). Lung IFNγ responses have been described as a surrogate marker

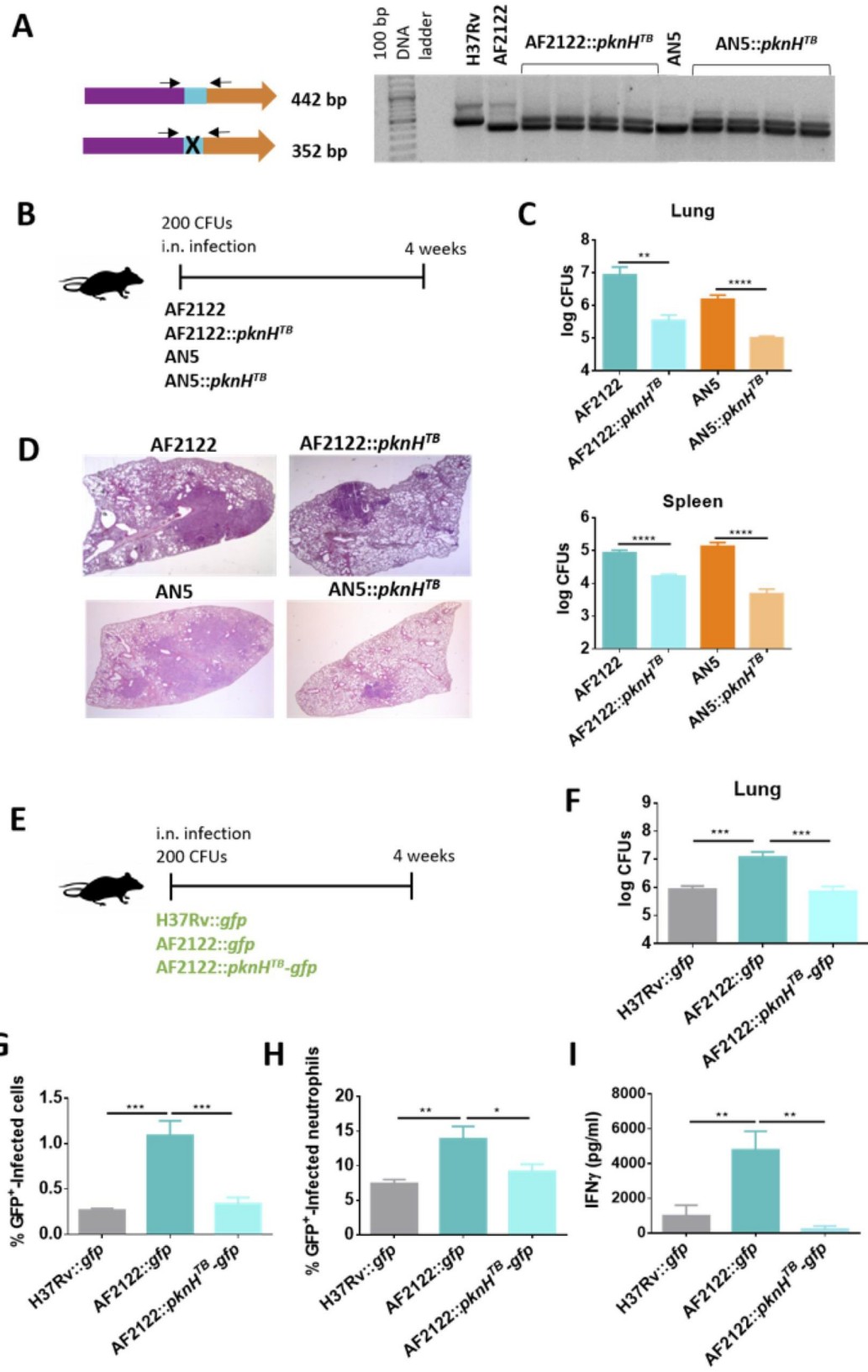

**Fig 3. Introduction of *M. tuberculosis pknH* into *M. bovis* strains reduces its virulence in the mouse model.** (A) Verification of *M. tuberculosis pknH* integration into *M. bovis::pknH*$^{TB}$ strains genome through colony PCR using two primer pairs flanking the deletion within the proline-rich region of *pknH* gene. (B) Schematic representation of the infection model in C57BL/6 mice. (C) Bacterial burden in lung and spleen from *M. bovis* AF2122 and AN5 WT and *M. bovis* AF2122:: *pknH*$^{TB}$ and AN5::*pknH*$^{TB}$ strains one month post-infection. (D) Representative images (1.5x) of heamtoxilin-eosin staining of lungs from mice infected with the different pathogens at one and four months post-infection. (E) Schematic representation of the infection model in C57BL/6 mice with fluorescent *M. bovis* and *M. tuberculosis* strains. (F) Bacterial burden in lung from *M. tuberculosis* H37Rv, *M. bovis* AF2122 and *M. bovis* AF2122::*pknH*$^{TB}$ strains one month post-infection. (G) Percentages of GFP+-infected cells from lung cell suspensions of *M. tuberculosis* H37Rv, *M. bovis* AF2122 and *M. bovis* AF2122::*pknH*$^{TB}$ at one month post-infection. (H) Percentages of GFP+-infected neutrophils from lung cell suspensions of *M. tuberculosis* H37Rv, *M. bovis* AF2122 and *M. bovis* AF2122::*pknH*$^{TB}$ at one month post-infection. (I) Specific IFNg response in lung single-cell suspensions from *M. tuberculosis* H37Rv and *M. bovis* AF2122 and AF2122:: *pknH*$^{TB}$ stimulated with PPD. IFNg specific response was measured in the supernatants after 48 hours of incubation by ELISA. Represented values correspond to the difference between IFNγ production in non-stimulated vs PPD stimulated cells. Data are represented as mean ± SEM from two independent experiments (n = 6 mice/group). Statistical analysis was performed by one-way ANOVA (C, F, G, H, I). *p < 0.05; **p < 0.01; ***p < 0.001; ****p < 0.0001.

for worse disease progression in different animal models, including mice and non-human primates [27–30].

## Transcriptomic characterization of the AF2122::pknH$^{TB}$ strain

In order to elucidate the molecular role of PknH and gain insight into the differences observed *in vivo* between *M. bovis* WT and *pknH*$^{TB}$ knock-in strains, a transcriptomic analysis of *M. bovis* AF2122/97 vs AF2122::*pknH*$^{TB}$ was performed from RNA extractions of *in vitro* cultures under standard conditions. Differentially expressed (DE) genes were filtered using a cutoff of p-value (adjusted) = 0.05 and log2fold change > 1.5 or < -1.5. Using these parameters, 145 DE genes were detected; 54 of them were down-regulated in AF2122::*pknH*$^{TB}$ compared to the WT AF2122 strain, and 91 of them were up-regulated (Fig 4A and S2 Fig). The DE genes found in this analysis are involved in a range of cellular processes. Genes involved in regulatory pathways, intermediate and lipid metabolism, cell wall processes and PE/PPE proteins were found to be differentially expressed, as well as 46 genes encoding conserved hypothetical proteins (Fig 4B and S2 Fig). As expected, one of the up-regulated genes in AF2122::*pknH*$^{TB}$ is *embR* (*Mb1298C*), a verified PknH substrate that is phosphorylated by PknH and that plays a role in the regulation of cell wall component synthesis. The *embCAB* operon, controlled by EmbR, also showed low level upregulation but this did not meet the statistical thresholds for inclusion in our DE genes. Also amongst the genes that were upregulated in AF2122::*pknH*$^{TB}$ were *espACD*, required for ESX-1 Type VII secretion (Fig 4C). Other genes involved in ESX-1 secretion did not show differential expression. The *espACD* locus is adjacent to the RD8 deletion in *M. bovis* [31], a deletion that has been shown to lead to altered regulation of *espACD* in *M. bovis* as compared to *M. tuberculosis* [32, 33].

Previous work has shown a role for PknH in the synthesis of phthiocerol dimycocerosates (PDIM). Deletion of *pknH* from *M. tuberculosis* was shown to decrease PDIM levels, with the PpsE polyketide synthase, part of the biosynthetic pathway, showing increased protein levels in the Δ*pknH* mutant [34]. However, none of the PDIM biosynthetic genes (*ppsABCDE*, *papA5*, *mas*, *fadD28*) were DE between AF2122::*pknH*$^{TB}$ and WT.

Among the identified DE genes which may have a role in virulence are *cysN* (*Mb1317*) and *cysD* (*Mb1316*), which were down-regulated in AF2122::*pknH*$^{TB}$. These genes are involved in sulfur metabolism and appear to play an important *in vivo* role [35]. Similarly, *mce* genes (Mb0178/mce1D, Mb0177/mce1C, Mb0176/mce1B and Mb0175/mce1A) were up-regulated in AF2122::*pknH*$^{TB}$ compared to the WT strain (Fig 4C). Proteins encoded by *mce* genes have been described to interfere with host cell signaling and it has been shown that mutation of

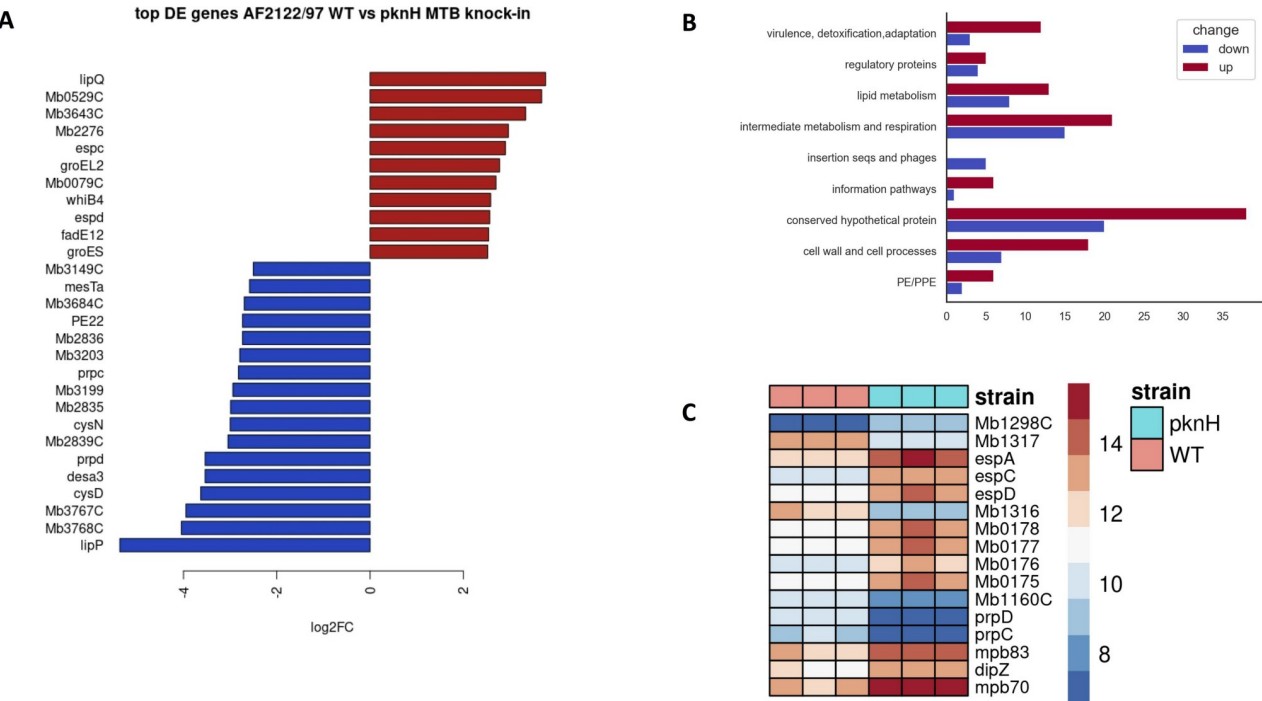

**Fig 4. RNA-seq analysis of *M. bovis* AF2122 wild type (WT) and *M. bovis* AF2122::*pknH^TB*.** (A) Top differentially expressed genes between WT and AF2122::*pknH^TB* on log2 scale. (B) Functional classification of genes showing differential expression (DE) between WT and AF2122::*pknH^TB* complement; functional classification as per Cole et al (1998). (C) Heat map of selected genes between WT and AF2122::*pknH^TB* (pknH) as referred to in text. Relative expression levels for each biological replicate are shown.

these genes affects *M. tuberculosis* virulence in mice [36, 37]. Also, the expression of two metabolic genes and their regulator namely *prpD*, *prpC* and *prpR* (enoding methylcitrate dehydratase, methylcitrate synthase and their transcriptional regulator, respectively; Fig 4C) were decreased in AF2122::*pknH^TB* relative to WT.

A comprehensive listing of DE genes between *M. bovis* AF2122/97 and *M. tuberculosis* H37Rv during in vitro growth has been previously described [38]. We therefore examined whether introduction of *pknH^TB* into *M. bovis* switched the expression of these DE genes in *M. bovis* to more closely match their expression in *M. tuberculosis* H37Rv. This revealed a minimal overlap, with only *Mb1904*, *Mb3504* and *PE31* showing expression in AF2122::*pknH^TB* that more closely resembled levels in *M. tuberculosis* H37Rv than *M. bovis* WT (S2 Table). There were also a subset of DE genes in AF2122::*pknH^TB* that amplified previously described DE genes between *M. bovis* AF2122 and *M. tuberculosis* H37Rv. For example, increased expression of *mpb83-dipz-mpb70* was seen in AF2122::*pknH^TB* (Fig 4C), a locus that is highly expressed in *M. bovis* as compared to *M. tuberculosis* [38, 39].

## Discussion

Understanding the molecular basis of pathogenesis of the bovine and human tubercle bacilli is crucial to define host tropism and to develop new approaches to control bovine TB. *M. bovis* has been described as being more virulent than *M. tuberculosis* in several animal models as well as having a wide host range that includes humans [3, 8–10]. Conversely, although *M. tuberculosis* is a successful human pathogen, it has been described to be attenuated in cattle [6, 7]. These major changes in host tropism and virulence belie the low levels of genetic diversity

within MTBC species. However, previous results in the literature provide evidence for how genomic differences between *M. bovis* and *M. tuberculosis* can lead to differential phenotypic features. For instance, evolutionary conserved mutations in the *M. bovis* PhoPR system were shown to explain the lack of trehalose-containing glycolipids in *M. bovis* compared to *M. tuberculosis* and have been associated with host preference [32]. Other genetic polymorphisms that define host specificity and infection phenotypes of *M. bovis* and *M. tuberculosis* remain to be elucidated. Here we revisited *M. tuberculosis* vs *M. bovis* infection in the mouse model and explored the role of RD900 locus and *pknH* genes in the context of MTBC evolution, and particularly the potential impact of RD900 on *M. bovis* and *M. tuberculosis* virulence phenotypes.

Since PknH has been described to regulate several physiological processes in *M. tuberculosis* and to have a role in virulence, variations across this genomic region may have major implications on the bacterial infection phenotype. The presence or absence of the RD900 region in MTBC species has been controversial, as it was first described as a lineage specific locus in an *M. africanum* strain, but it was later found to be also present in *M. bovis* AF2122/97 [17, 18, 21]. RD900 contains two homologous *pknH* genes flanking the *tbd2* gene. Recombination between *pknH* genes in some MTBC species results in the loss of the *tbd2* gene and in the generation of a single *pknH* gene. The presence of homologous and repetitive sequences within flanking *pknH* genes can lead to errors in genome sequencing and annotation when using older MTBC genome assemblies as a reference, and genome annotations could contain errors within the RD900 region, similar to that seen in the original *M. bovis* AF2122/97 genome assembly [40]. Therefore, we aimed to study RD900 variation across MTBC species not only because of the potential importance of this locus in defining bacterial phenotypes, but also to clarify the configuration of this region across MTBC species.

Our genomic analysis revealed the presence of the RD900 locus in *M. canetti*, *M. tuberculosis* 'ancient' lineages, *M. africanum*, *M. mungi*, *M. microti*, *M. pinnipedii*, *M. caprae* and *M. bovis* strains. However, variation in the presence or absence of RD900 was found in *M. tuberculosis* Lineage 1, *M. caprae* and *M. bovis* strains, including BCG strains. These results suggest that deletion of RD900 occurred independently in different lineages and different strains within these lineages. The occurrence of independent recombination events between *pknH* paralogues at different evolutionary branches in RD900 deleted (RD900-) strains could result in the generation of a truncated *pknH* gene. However, variant calling analysis of the resulting *pknH* gene in RD900- strains revealed a single intact *pknH* gene with no frameshift mutations, suggesting that independent homologous recombination events generate a potentially functional gene. This could indicate a possible selective advantage for the reduction of *pknH* paralogues or a potential need to conserve at least one functional *pknH* gene. Interestingly, similar recombination patterns and genomic organization of *pknH* genes has also been found in other *Mycobacterium* species such as *M. abscessus* and *M. marinum* [17], and similar recombination-deletion events between homologous genes that result in the formation of a single new gene have also been described in cases such as *pks5* in *M. canettii* strains. Recombination of *pks5* genes and loss of *pap* seems to have occurred in the common ancestor of the MTBC and is proposed to have contributed to evolutionary success and host adaptation, as it is related with the loss of lipooligosaccharide production in MTBC species [41]. In addition to the absence/presence of RD900 that results in one or two *pknH* genes being present, another major variation within the *pknH* gene is the presence of a proline-rich region that is attached to the catalytic domain of the PknH protein. This proline-rich region is deleted in the *pknH2* gene of all the analysed MTBC isolates that contain a complete RD900 locus. Furthermore, all of the analysed *M. bovis* and *M. caprae* strains have the proline-rich region deleted in both *pknH1* and *pknH2* genes in RD900+ strains. Consequently, the single *pknH* gene resulting from *pknH* recombination events in *M. bovis* and *M. caprae* RD900- strains carries the same deletion.

**Fig 5. Hypothetical RD900 homologous recombination events in *M. tuberculosis* "modern" lineages and *M. bovis* and *M. caprae*.**

Considering the *M. canettii* RD900 locus as the ancestral state of the RD900 region, and based on sequence homology between the different domains of PknH1 and PknH2, it can be hypothesized that the recombination site in the *M. tuberculosis* "modern" lineages could be located at the homologous kinase domains, resulting in an in-frame *pknH* gene composed of the kinase domain from *pknH2* and the proline-rich region, transmembrane and sensor domain from *pknH1*. In *M. bovis* and *M. caprae* strains with the complete RD900 locus, a deletion in the proline-rich domain may have happened first, resulting in an RD900 locus with both *pknH* genes carrying the same deletion. Hence, in *M. bovis* and *M. caprae* strains with a deleted RD900 locus, homologous recombination between the two *pknH* genes would result in a single *pknH* lacking the proline-rich region (Fig 5).

In addition to *pknH2*, RD900-negative MTBC strains lack in the *tbd2* gene. The specific function of this gene is unknown, as well as the consequences of its absence in the RD900 negative strains. *tbd2* gene encodes a transmembrane ATP binding cassette (ABC) transport protein of unknown function that has a central ATP binding domain and six possible membrane-spanning domains in the C-terminal portion [17]. The N-terminal region contains a FHA domain, which is a phosphoprotein recognition domain. Thus, it can be hypothesized that Tbd2 could be phosphorylated by the adjacent PknH protein or by others STPKs. Indeed, phosphorylation of other PknH substrates, such as EmbR, has been shown to be mediated by the FHA domain [12].

The genomic analysis of the RD900 locus presents some limitations. Since genome sequencing reads were aligned to a reference sequence, it was not possible to detect large insertions or deletions not present in the reference sequence. Furthermore, the sequence identity between *pknH1* and *pknH2* leads to multi-mapping reads, which are reads that can map to more than one locus within the reference sequence and that complicates definitive assignment of read positions. Another approach to analyse this region would be to do *de novo* assembly of the genome and to align the contigs to a reference sequence. However, the issue with this latter method is that the sequence of the RD900 locus was split into different contigs if the available read length was lower than 150 bp, which was the case for the most of the available genomic data that was analysed in this study.

We characterized infection and virulence differences between *M. tuberculosis*, *M. bovis* and *M. africanum* strains in C57BL/6 mice. In agreement with previous literature, we found *M. bovis* to be more virulent than *M. tuberculosis* during early and advanced stages of infection in the mouse model, showing greater lung pathology and higher bacterial burden in lungs and other organs, especially the liver and kidneys. This phenotype correlates with a higher percentage of infected lung cells in *M. bovis* infected mice one month post-infection. At this time point, the majority of infected lung cells were neutrophils and a minority were macrophages. The level of infected neutrophils was higher in *M. bovis* infected mice, which also correlates with higher neutrophil infiltration in the total lung cell population. Increased neutrophilia has been previously related to exacerbation of TB pathology and poor prognosis, and a neutrophil-driven IFN-inducible transcript signature in human whole blood of patients with active TB correlates with clinical severity [23, 25, 26, 42]. Here we show that lung neutrophilia and increased numbers of infected neutrophils in the lung correlate with enhanced pathology, higher bacterial burden and dissemination in *M. bovis* infected mice compared to mice infected with *M. tuberculosis* or with the closely related *M. africanum*.

Our PCR analysis of *M. bovis* cDNA using proline-rich region flanking primers indicated that *pknH* gene is expressed by *M. bovis* strains (S1B Fig). Indeed, RNAseq analysis from AF2122 strain showed reads from both *pknH* copies [43], suggesting that deletion of the proline-rich region in *M. bovis* and *M. caprae* PknH could lead to generation of an uncomplete PknH protein, which may contribute *to M. bovis* and *M. caprae* virulence. This hypothesis would be consistent with a previous observation from the literature describing increased virulence in a *pknH*-deficient *M. tuberculosis* strain. The proline-rich region in PknH is not present in other mycobacterial STPKs and its function in PknH its unknown. It could act as a linker between the catalytic and transmembrane domains, or it may be involved in protein-protein interactions or adaptor functions, as similar proline-rich domains in eukaryotic signalling proteins have been associated with functions in protein recognition, binding and oligomerization [44–46].

To address the possible impact of PknH differences between *M. tuberculosis* and *M. bovis*, two *M. bovis* strains carrying the *pknH* gene from *M. tuberculosis* H37Rv were constructed, and their virulence phenotypes were studied in the mouse model. A reduction in lung pathology and bacterial load in lungs and spleen was observed for both *M. bovis* AN5 and AF2122 *pknH*$^{TB}$ knock-in strains compared to their parental *M. bovis* WT strains. This phenotype was further confirmed in AF2122 strains using GFP-expressing bacteria. A reduction in lung infected cells and infected neutrophils in the AF2122::*pknH*$^{TB}$ strain compared to the WT AF2122 was observed, which correlates with lower bacterial load and lung pathology and further supports a detrimental role for neutrophils in pulmonary TB infection.

In order to better understand the molecular processes that are regulated by PknH, a transcriptomic analysis of *M. bovis* AF2122/97 WT and AF2122::*pknH*$^{TB}$ strains was performed that revealed 145 DE genes under standard culture conditions. The gene encoding EmbR, a verified PknH substrate, was found to be up-regulated in the AF2122::*pknH*$^{TB}$ strain. EmbR is a transcriptional regulator that controls the synthesis AG and LAM, which are structural components of mycobacterial cell wall that may have a role in the modulation of host response during infection [12, 14].

The upregulation of the *espACD* locus in the *M. bovis* AF2122::*pknH*$^{TB}$ may seem at odds with the decreased virulence of this recombinant strain. The *espACD* has been shown to be essential in ESX-1 secretion, a key determinant of *M. tuberculosis* virulence [32, 47]. In PhoPR mutants, it has been shown that reduced *espACD* expression leads to reduced ESX-1 secretion and reduced virulence (34). However, it has also been previously shown that upregulation of *espACD* does not always lead to increased ESX-1 secretion or increased virulence. Pang et al

[48] showed that while deletion of MprAB in *M. tuberculosis* led to increased transcription of *espA* and *espD*, there was a concomitant decrease in EspA, EspB and EsxA secretion. Furthermore, the *mprAB* mutant induced decreased levels of IL1-β and TNF-α during macrophage infections. Further exploration of the linkages between PknH and the ESX-1 system in *M. bovis* will be needed to clarify whether a similar situation as seen by Pang et al pertains. As noted above, the deletion of RD8 leads to altered regulation of the *espACD* locus in *M.bovis* [32, 33], so PknH may play a distinct role in regulation of this locus in *M. bovis* as compared to *M. tuberculosis*.

As introduction of the *pknH*$^{TB}$ allele into *M. bovis* significantly altered gene expression, the question arose as to whether this altered transcriptome would recapitulate some of the known transcriptomic differences between *M. bovis* and *M. tuberculosis* [38]. While a subset of genes switched expression in AF2122::*pknH*$^{TB}$ to resemble expression in *M. tuberculosis* (S2 Table), they did not provide clear clues as to the altered virulence of the recombinant.

A variety of genes involved in intermediate and lipid metabolism were found to be DE between the *M. bovis* WT and *pknH*$^{TB}$ knock-in strain, which may lead to differences in metabolic pathways or lipid profiles that could also contribute to virulence differences. Interestingly, *cysD* and *cysN* genes were found to be down-regulated in AF2122::*pknH*$^{TB}$ strain. These two genes are involved in sulfur metabolism and have been found to be highly expressed in intracellular conditions as well as in bacteria isolated from mouse lungs [35], suggesting that their repression in the AF2122::*pknH*$^{TB}$ strain may also play a role in its attenuation. The downregulation of the methylcitrate cycle genes *prpCD* (and their regulator, *prpP*) may suggest a decrease in lipid catabolism which is a key *in vivo* carbon source; nevertheless, previous work showed that a Δ*prpDC M. tuberculosis* mutant was not attenuated in a mouse model [49], hence arguing against a role for the downregulation of these genes in attenuation of AF2122::*pknH*$^{TB}$. Additionally, while a large number of genes encoding hypothetical and PE/PPE proteins were found to be DE, their function and hence possible role in virulence remains to be elucidated.

PknH is a STPK master regulator that can phosphorylate and regulate other mycobacterial STPKs, which in turn control other substrates and bacterial functions [50]. Thus, a complex interplay between multiple factors and cross-regulated pathways is likely involved in the gene expression profile and ultimate phenotype of the PknH mutant strains. Additionally, culture conditions may also be an important factor for the identification of PknH-regulated genes and for the ultimate determination of the AF2122 WT vs AF2122::*pknH*$^{TB}$ gene expression profile. PknH has been described to phosphorylate DosR, which is involved in the regulation of the latency response under stress conditions [51]. PknH phosphorylation of DosR enhances its DNA binding activity and favours the induction of DosR-regulated genes, whose expression is altered in an *M. tuberculosis* H37RvΔ*pknH* mutant under stress culture conditions [52]. DosR and DosR-regulated genes were not identified as DE genes in our RNA-seq analysis of AF2122 vs AF2122::*pknH*$^{TB}$. However, it is of course possible that these regulons could be affected by the intracellular conditions during *in vivo* infection with these strains. Therefore, although a variety of genes were found to be DE between these strains in basal *in vitro* conditions, further studies of expression differences using other culture conditions that regulate PknH induction, such as stress conditions (hypoxia or NO), would be useful to provide a more complete view of PknH regulation in these strains.

Previous studies have demonstrated higher virulence of *M. bovis* compared to *M. tuberculosis* in several animal models, such as mice or rabbits. In addition, experimental infections of cattle have demonstrated that *M. tuberculosis*, at least from lineage 4, is attenuated in the bovine host [6, 7]. Clearly, as we show here, the genetic differences between *M. bovis* and *M. tuberculosis* can contribute to the varying degrees of virulence shown for these animal hosts.

Ultimately however, an evolutionary successful strategy depends on host factors as well as pathogen virulence, the interaction of which drives transmission and maintenance of the pathogen in the host population. Indeed, *M. bovis* is able to spread and sustain across different animal populations, both domesticated (e.g. cattle) and wild (e.g. badgers and deer), yet human-to-human transmission of *M. bovis* is a rare event. Thus, one could hypothesize that *M. bovis* has evolved to exploit dissemination routes distinct from respiratory transmission. Conversely, *M. tuberculosis* has evolved towards human adaptation to which respiratory dissemination is integral and which may rely on a functional PknH.

In relation to the evolution of the distinct human vs animal adapted MTBC, the two component PhoPR system seems to have a crucial role. *M. bovis* strains contain point mutations in the region encoding the extracellular domain of PhoR, which impairs the signalling of this system. Thus, as described by Gonzalo-Asensio et al (34), the expression of *phoP* regulon genes is divergent in *M. bovis* as compared to *M. tuberculosis*, which could again play a role in their compromised capacity to spread via human-to-human. Interestingly, an *M. bovis* strain, called the 'B strain' caused a severe TB outbreak in Spain in the 1990s and disseminated efficiently among humans. This latter strain overexpressed the *phoP* gene and subsequently expressed the PhoP regulon at levels similar to *M. tuberculosis*, highting the linkages between this regulon and efficient human-to-human transmission.

In summary, our results suggest a role for PknH in the differential regulation of virulence and infection in *M. bovis* and *M. tuberculosis*. The apparent alteration, or lack, of PknH activity in *M. bovis* leads to increased virulence in animal hosts. Indeed, this finding may help explain the reduced evidence for latency in *M. bovis* infection in animals, whereby infected livestock or wild mammals usually exhibit a more progressive and disseminated disease presentation as compared to human disease [53]. Our findings therefore have implications for our understanding of the genetic underpinnings and evolution of host adaptation in the MTBC.

## Materials and methods

### Ethics statement

Experimental work was conducted in agreement with the Spanish Policy for Animal Protection RD53/2013 and the European Union Directive 2010/63 for the protection of animals used for experimental and other scientific purposes and experimental procedures were approved by the Ethics Committee for Animal Experiments of University of Zaragoza (CEA).

### Animals

C57BL/6JR mice were purchased from Janvier Biolabs. All mice were housed and maintained in specific pathogen-free conditions and observed for any sign of disease in the facilities of Centro de Investigación de Encefalopatías y Enfermedades Transmisibles Emergentes (ES 50 297 0012 009). Male and female mice between the ages of 8 to 10 weeks were used for all the experiments.

### Bacterial strains and culture conditions

Mycobacterial strains were grown at 37°C in Middlebrook 7H9 broth (BD Difco) liquid medium supplemented with 0.05% Tween 80 (Sigma) and 10% (v/v) Middlebrook albumin dextrose catalase enrichment (ADC; BD Biosciences 0.2% dextrose, 0.5% bovine serum albumin, 0.085% NaCl and 0.0003% beef catalase). For cultures in solid media, Middlebrook 7H10 agar (BD Difco) supplemented with 10% ADC (BD Biosciences) was used. When required, medium was supplemented with 20 μg/ml of kanamycin (Km) or 20 μg/ml of streptomycin

(Sm). For the culture of *M. bovis* strains liquid and solid media was supplemented with sodium pyruvate (Sigma) 4 mg/ml.

All GFP-expressing carry with the replicative plasmid pJKD6 encoding the green fluorescent protein (GFP), a kind gift from Luciana Leite, Butantan Institute, Brazil. This plasmid carries de *gfp* gene under the control of a strong promoter generated through error-prone PCR of the PL5 promoter from the mycobacteriophage L5 resulting in high levels of GFP expression [54].

Bacterial suspensions for intranasal infections were prepared in PBS from quantified glycerol stock solutions.

## Construction of *M. bovis*::*pknH^{TB}* knock-in strains

For the construction of the *M. bovis*::*pknH^{TB}* knock-in strains, the *pknH* gene from *M. tuberculosis* H37Rv (*pknH^{TB}*) was cloned using the integrative plasmid pMV361. The gene insert was synthesized and subcloned in NheI site of pMV361 by Genscript Company (Piscataway, USA). *M. bovis* AF2122/97 and AN5 competent cells were obtained and electroporated with pMV361-*pknH^{TB}* plasmid. Oligonucleotides used for PCR verifications are detailed in S3 Table.

GFP-expressing strains of *M. tuberculosis* H37Rv and Mt103, *M. bovis* AF2122/97 and *M. africanum* HCU2828 were generated by transformation by electroporation with the replicative plasmid pJKD6 carrying a kanamycin resistance cassette. GFP-expressing AF2122::*pknH^{TB}* was generated through transformation by electroporation with the replicative plasmid pJKD6 carrying a streptomycin resistance cassette.

## Intranasal infection

Mice were infected with a low dose ($\approx$100–200 CFUs) of the different mycobacterial strains by the intranasal route. The animals were anesthetized by inhalation route with Isofluorane (Isboa Vet) using a vaporizer and intranasal administration was performed with two instillations of 20 μl of the bacterial suspension prepared in PBS. Bacterial suspensions for infection were plated in solid agar medium to determine the CFUs used for *in vivo* challenges.

## Bacterial burden determination

For bacterial burden determination in lung, spleen, liver and kidneys, the organs were aseptically removed and homogenized in 1 ml of $H_2O$ using a GentleMacs dissociator (Miltenyi Biotec). CFUs were determined by plating serial dilutions on solid 7H10 medium supplemented with 10% ADC and sodium pyruvate when required.

## Histological analysis

For lung histopathology analysis, left lung was removed and fixed in 4% formaldehyde for 24 hours prior to hematoxylin-eosin (HE) staining. Histological staining was performed in the Pathological Anatomy Service from CIBA (Zaragoza, Spain) and images were obtained with a Leica DM5000B optical microscope.

## Lung single cell suspensions

For lung cell suspensions, lungs were aseptically removed and homogenized in HEPES buffer (HEPES 10 mM, NaCl, 150 mM, KCl 5 mM, MgCl2 1 mM, CaCl2 1,8 mM pH 7,4) containing DNaseI (AppliChem) 40 IU/ml and Collagenase D (Roche) 2 mg/ml using a GentleMacs dissociator (Miltenyi Biotec) according to manufacturer instructions. Lungs were incubated at 37˚C for 30 minutes and further homogenized with the GentleMacs dissociator. The

homogenates were filtered through a 70 μm cell strainer (MACS SmartStainers, Miltenyi Biotec) and centrifuge at 1500 rpm for 5 min. Red blood cells were lysed using Red Blood Cell Lysing Buffer (Sigma-Aldrich) and finally, cells were pelleted and resuspended in RPMI (Gibco) culture medium.

### Antibody staining and flow cytometry analysis

Lung single cell suspensions were plated in U-bottom 96-well plates and incubated for 15 minutes at 4˚C with FcR blocking reagent (Miltenyi Biotec). Surface staining was performed for 20 min at 4˚ C using different combinations of antibodies to define lung myeloid populations of macrophages and neutrophils. Then, cells were washed and fixed with 4% paraformaldehyde for 30 minutes.

Cells were acquired using a Gallios flow cytometer (Beckman Coulter) and the results were analysed using Weasel Software.

For the analysis of lung myeloid populations, total lung macrophages were defined as CD45 +Ly6G-CD11c+ and neutrophils as CD45+Ly6G+CD11c-CD11b+ (S3 Fig).

### Cytokine determination

Lung cell suspensions were plated in U-bottom 96-well plates and incubated for 48 h in the presence or absence of PPD 10 μg/ml. Supernatants were collected to determine IFNγ response. Quantification of IFNγ was performed using a specific commercial ELISA kit (Mabtech Biotech) according to manufacturer instructions.

### Statistics

Mice were distributed in groups of at least 6 animals per cage prior to experimental procedures. Results were not blinded for analysis and randomization was not applicable to these studies. Rstudio and GraphPad Prism software (version 6) was used for graphical representation and statistical analysis. Statistical test used in each experiment are indicated in the figure legends. Outlier values were determined applying Grubb's test. A p value of $< 0.05$ was considered significant ($^*p < 0.05$, $^{**}p < 0.01$, $^{***}p < 0.001$, $^{****}p < 0.0001$).

### Comparative genomics

Computational analysis of RD900 region across MTBC was performed using WGS data of 60 different MTBC isolates (S1 Table). We did not use annotations based on existing assemblies since misannotations are likely present in this locus due to the fact that homologous domains shared between *pknH* genes cannot be uniquely aligned with short reads. Such reads are marked as multi-mappers and ignored; resulting assemblies are therefore split into contigs over such a region and annotation software may miss genes. To avoid this, we downloaded the original raw sequence reads in fastq format from GenBank for each available isolate (S1 Table).

The raw reads were aligned to a reference sequence (*M. africanum* GM041182 RD900 locus) representing the most complete version of the RD900 across the MTBC. In this way we could most readily see missing elements in the other species. Reads were aligned using bwa (mem algorithm) [55] and the resulting bam files were sorted and indexed using samtools. The alignments were visualized in IGV [56] to identify the presence or absence of RD900 and the presence of the deletion in the proline-rich region within *pknH* genes. Finally, a variant calling analysis was performed to identify point mutations present in this region from the different

MTBC analysed species. Variant calling, filtering and consequence calling were performed using freebayes (https://arxiv.org/abs/1207.3907) and bcftools.

## RNA-seq analysis

*M. bovis* AF2122 wild type and *M. bovis* AF2122::pknHMtb knock-in were grown in triplicate standard 7H9 medium with pyruvate (see Bacterial strains and culture conditions above) to mid-log phase (OD0.4–0.6). RNA was extracted by QIAzol lysis Reagent and RNeasy Mini Kit (Qiagen). DNA was removed by TURBO DNase (Thermo Fisher) and RNA Clean-Up and Concentrator (Zymo).

The concentration and quality of all the RNA samples were checked using Nanodrop, Qubit (Thermo Fisher) and Agilent 2100 bioanalyzers. RNA was sent for commercial sequencing (Novogene) for library preparation and 150 bp paired-end reads sequencing on an Illumina HiSeq.

RNA-seq data were first processed using FastQC (https://www.bioinformatics.babraham. ac.uk/projects/fastqc/) tool to check the quality of the raw reads. The raw reads were trimmed by Trim-galore (https://www.bioinformatics.babraham.ac.uk/projects/trim_galore/) to trim off reads with bad quality and adapters used for sequencing. Alignment to the reference genome *M. bovis* AF2122/97 was done using bwa mem [55]. Gene expression was determined from the resulting bam files using featureCounts [57]. The count data was then analysed for differential expression of genes between the wild type and mutant with the DESeq2 R package [58]. Significant genes were selected at a log2 fold change threshold of 2.

## Supporting information

**S1 Fig. Expression of *pknH*$^{TB}$ in *M. bovis*::*pknH*$^{TB}$ knock-in strains.** (A, B) Confirmation of *pknH*$^{TB}$ presence in *M. bovis*::*pknH*$^{TB}$ strains. cDNA was analyzed by PCR, either with the reverse primer included inside the proline-rich region (A), or with both primers flanking this sequence (B). In the first case, cDNA amplification was only observed when *pknH*$^{TB}$ was expressed, whereas in the second case amplification occurred in both cases, with a different fragment size depending on the pknH expressed. (C) *pknH*$^{TB}$ expression was measured by qRT-PCR using the proline-rich region flanking primers. Graph represents mean±SD from one experiment with three replicates, corresponding to the fold-change value in comparison to H37Rv expression.
(TIF)

**S2 Fig. Functional classification of differentially-expressed (DE) genes between AF2122 wild-type and *pknH*$^{TB}$ knock-in strains.**
(TIF)

**S3 Fig. Flow cytometry analysis for the identification of lung macrophages and neutrophil populations.**
(TIF)

**S1 Table. Genomic data of MTBC isolates used for the genomic analysis of RD900 locus.**
(DOCX)

**S2 Table. Filtered DE gene overlap between *M.bovis*::pknH$^{TB}$ vs *M. bovis* WT, and *M. bovis* WT vs *M. tuberculosis*.** *Gene expression data from *M. tuberculosis* were obtained from Malone *et al.* [38].
(DOCX)

**S3 Table. Primer sequences used for PCR and qRT-PCR.**
(DOCX)

## Acknowledgments

The authors acknowledge the Scientific and Technical Services from Instituto Aragonés de Ciencias de la Salud and Universidad de Zaragoza for their assistance. We thank Jeff Chen for advice and discussion.

## Author Contributions

**Conceptualization:** Carlos Martin, Stephen V. Gordon, Nacho Aguilo.

**Data curation:** Elena Mata, Damien Farrell, Ruoyao Ma.

**Formal analysis:** Elena Mata, Damien Farrell, Stephen V. Gordon, Nacho Aguilo.

**Funding acquisition:** Alberto Anel, Carlos Martin, Stephen V. Gordon, Nacho Aguilo.

**Investigation:** Elena Mata, Damien Farrell, Ruoyao Ma, Santiago Uranga, Ana Belen Gomez.

**Methodology:** Elena Mata, Nacho Aguilo.

**Project administration:** Stephen V. Gordon, Nacho Aguilo.

**Resources:** Ruoyao Ma, Marta Monzon, Juan Badiola, Carlos Martin, Stephen V. Gordon, Nacho Aguilo.

**Software:** Elena Mata, Damien Farrell, Ruoyao Ma.

**Supervision:** Alberto Anel, Carlos Martin, Stephen V. Gordon, Nacho Aguilo.

**Validation:** Elena Mata.

**Visualization:** Elena Mata, Stephen V. Gordon, Nacho Aguilo.

**Writing – original draft:** Elena Mata, Alberto Anel, Jesús Gonzalo-Asensio, Carlos Martin, Stephen V. Gordon, Nacho Aguilo.

**Writing – review & editing:** Elena Mata, Carlos Martin, Stephen V. Gordon, Nacho Aguilo.

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
