## [Decision Letter · Decision Letter 0]

25 Aug 2020

Dear Dr Aguilo,

Thank you very much for submitting your manuscript "Independent genomic polymorphisms in the PknH serine threonine kinase locus during evolution of the Mycobacteriumtuberculosis Complex affect virulence and host preference" for consideration at PLOS Pathogens. As with all papers reviewed by the journal, your manuscript was reviewed by members of the editorial board and by several independent reviewers. The reviewers appreciated the attention to an important topic. Based on the reviews, we are likely to accept this manuscript for publication, providing that you modify the manuscript according to the review recommendations.

Sincerely,

Marcel A. Behr

Associate Editor

PLOS Pathogens

JoAnne Flynn

Section Editor

PLOS Pathogens

Kasturi Haldar

Editor-in-Chief

PLOS Pathogens

orcid.org/0000-0001-5065-158X

Michael Malim

Editor-in-Chief

PLOS Pathogens

orcid.org/0000-0002-7699-2064

Reviewer Comments (if any, and for reference):

Reviewer's Responses to Questions

**Part I - Summary**

Reviewer #1: Overall, I find the paper of Mata et al. to be a relatively straightforward and cleanly conducted study that aims to clarify the potential role of the variable RD900 region in host specificity and virulence within the MTBC. As such, I have relatively few specific comments or suggestions to improve the manuscript.

In particular, I find the virulence phenotype very impressive and I really respect the author’s decision to utilize two separate M. bovis strains for their knock-in work. This helps to remove any niggling doubt I may have otherwise had regarding whether the loss of virulence may have been due to some other off-target effect (eg. the spontaneous loss of PDIM). The introduction of GFP into AF2122 with no apparent alteration of virulence also serves effectively as a “empty-vector” control.

Reviewer #2: In this manuscript, the authors analyze the sequences of a particular genomic region of difference across a set of more than 60 strains belonging to the different lineages of the M. tuberculosis complex (MTBC). The authors describe that this region, named RD900 is comprising one or two copies of a pknH gene encoding the serine/threonine protein kinase PknH. The authors hypothesize that this region is prone to frequent homologous recombination, leading to different genomic situations in several lineages or strains.

The authors further show in the pknH genes there is also a polymorphic region observed that is concerning a sequence encoding a proline rich section of the protein.

Based on these genomic analyses and on previous literature the authors hypothesize that the pknH region is involved in the virulence of MTBC members, and in particular in the enhanced virulence of M. bovis strains in the C57BL/6 mouse model compared to reference M. tuberculosis strains.

To follow-up this hypothesis, the authors have constructed knock-in variants of 2 different M. bovis strains that contain an integrated vector carrying the pknH gene of M. tuberculosis H37Rv.

The authors use these constructs to do mouse infection experiments and RNA seq. By this approach they firzst confirm that M. bovis wild-type strains show an increased virulence in the C57BL/6 mouse model compared to M. tuberculosis strains, and then find lower virulence of the recombinant M. bovis strains compared to the parental strains. In the RNA seq experiment of the parental and recombinant M. bovis strains they find a range of genes up (e.g. espACD) or down regulated cysD/N.

**Part II – Major Issues: Key Experiments Required for Acceptance**

Reviewer #1: (No Response)

Reviewer #2: Overall the manuscript is interesting as it deals with an important subkject, i.e. the virulence of MTBC strains. The genomic analysis is convincing and the working hypotheses are well taken.

The mouse infection experiments are also convincing as there is a > than 1 log CFU difference shown betweenis strains and M. tuberculosis strains, as well as between M. bovis parental strains and pknH knock in mutants. However, one point should be better explained by the authors, how they ascertained that the initial dose that was used to infect mice is comparable between strains, as there are no day 1 data shown.

For the discussion, it would be helpfull if the authors could better explain whether the M. bovis strains used in the mouse experiments expressed the two copies of pknH they contain, or if the two genes were not expressed ?. This was not clear from the reading and it would help to better understand the rational of adding the pknH gene from M. tuberculosis.

The RNA seq data should also be explained in more detail. It is not clear why an increased expression of espACD in the pknH TB knock in strains should cause decreased virulence, especially as the regulation of the espACD operon is complex in M. tuberculosis (PhoPR, EspR, etc) and different in M. bovis (RD8 etc). The role of pknH in this regulation process should be hypothesized.

**Part III – Minor Issues: Editorial and Data Presentation Modifications**

Reviewer #1: The are two areas where, in my opinion, the author’s may have gone a little further to strengthen their analysis:

1) Particularly where the author’s make the claim that there is variability within the same lineage regarding the structure of the pknH locus, it would have been preferable to have confirmed some of the gene arrangements via PCR and/or Sanger sequencing (or cloning and sequencing) rather than relying solely on alignments generated from someone else’s WGS data. As the author’s themselves have highlighted, recombination and repetitive sequences in this variable region have previously led to annotation errors (and confusion) for various MTBC members.

2) In regards to the RNAseq analysis comparing M. bovis with M. bovis::mtb-pknH (Fig. 4a) – are the author’s aware if any of the genes that appear in their list of the “top DE genes” are also variably expressed between, for example, M. canettii or M. tuberculosis when compared to M. bovis?

Reviewer #2: Minor issues. The manuscript should be proofread by a native English speaker, as there are several grammar mistakes apparent, e.g. line 463.

the nomenclature of pknHTB gene is confusing, "TB" could be noted as superscript or in a similar fashion.

a seperate paragraph with heading should be written for the RNA seq results.

It would be interesting to read the authors opinion on the role of the tbd2 gene present in RD900.

In reaction to a previous paper on the virulence of M. bovis and M. tuberculosis, interesting comments were written on the ability or inability of certain MTBC members to spread in certain hosts ( PMID: 25174642; PMID: 25435136) which might also apply to this work, and which should be discussed

PLOS authors have the option to publish the peer review history of their article (what does this mean?). If published, this will include your full peer review and any attached files.

Reviewer #1: No

Reviewer #2: No
---

## [Editor Report · Decision Letter 1]

9 Oct 2020

Dear Dr Aguilo,

We are pleased to inform you that your manuscript 'Independent genomic polymorphisms in the PknH serine threonine kinase locus during evolution of the Mycobacteriumtuberculosis Complex affect virulence and host preference' has been provisionally accepted for publication in PLOS Pathogens.

Best regards,

Marcel A. Behr

Associate Editor

PLOS Pathogens

JoAnne Flynn

Section Editor

PLOS Pathogens

Kasturi Haldar

Editor-in-Chief

PLOS Pathogens

orcid.org/0000-0001-5065-158X

Michael Malim

Editor-in-Chief

PLOS Pathogens

orcid.org/0000-0002-7699-2064
---

## [Editor Report · Acceptance letter]

1 Dec 2020

Dear Dr Aguilo,

We are delighted to inform you that your manuscript, "Independent genomic polymorphisms in the PknH serine threonine kinase locus during evolution of the Mycobacteriumtuberculosis Complex affect virulence and host preference," has been formally accepted for publication in PLOS Pathogens.

Best regards,

Kasturi Haldar

Editor-in-Chief

PLOS Pathogens

orcid.org/0000-0001-5065-158X

Michael Malim

Editor-in-Chief

PLOS Pathogens

orcid.org/0000-0002-7699-2064